# Screening of Suitable Reference Genes for Immune Gene Expression Analysis Stimulated by *Vibrio anguillarum* and Copper Ions in Chinese Mitten Crab (*Eriocheir sinensis*)

**DOI:** 10.3390/genes14051099

**Published:** 2023-05-17

**Authors:** Fengyuan Yan, Hui Li, Xue Chen, Junjie Yu, Shengyan Su, Jianlin Li, Wei Ye, Yongkai Tang

**Affiliations:** 1National Demonstration Center for Experimental Fisheries Science Education, Shanghai Ocean University, Shanghai 201306, China; 2Wuxi Fisheries College, Nanjing Agricultural University, Wuxi 214081, China; 3Key Laboratory of Freshwater Fisheries and Germplasm Resources Utilization, Ministry of Agriculture and Rural Affairs, Freshwater Fisheries Research Center, Chinese Academy of Fishery Sciences, Wuxi 214081, China

**Keywords:** Chinese mitten crab, reference gene, qRT-PCR, *Vibrio anguillarum*, copper ion

## Abstract

The reference gene expression is not always stable under different experimental conditions, and screening of suitable reference genes is a prerequisite in quantitative real-time polymerase chain reaction (qRT-PCR). In this study, we investigated gene selection, and the most stable reference gene for the Chinese mitten crab (*Eriocheir sinensis*) was screened under the stimulation of *Vibrio anguillarum* and copper ions, respectively. Ten candidate reference genes were selected, including arginine kinase (*AK*), ubiquitin-conjugating enzyme E2b (*UBE*), glutathione S-transferase (*GST*), glyceraldehyde-3-phosphate dehydrogenase (*GAPDH*), elongation factor 1α (*EF-1α*), α-tubulin (*α-TUB*), heat shock protein 90 (*HSP90*), β-actin (*β-ACTIN*), elongation factor 2 (*EF-2*) and phosphoglucomutase 2 (*PGM2*). Expression levels of these reference genes were detected under the stimulation of *V. anguillarum* at different times (0 h, 6 h, 12 h, 24 h, 48 h and 72 h) and copper ions in different concentrations (11.08 mg/L, 2.77 mg/L, 0.69 mg/L and 0.17 mg/L). Four types of analytical software, namely geNorm, BestKeeper, NormFinder and Ref-Finder, were applied to evaluate the reference gene stability. The results showed that the stability of the 10 candidate reference genes was in the following order: *AK* > *EF-1α* > *α-TUB* > *GAPDH* > *UBE* > *β-ACTIN* > *EF-2* > *PGM2* > *GST* > *HSP90* under *V. anguillarum* stimulation. It was *GAPDH* > *β-ACTIN* > *α-TUB* > *PGM2* > *EF-1α* > *EF-2* > *AK* > *GST* > *UBE* > *HSP90* under copper ion stimulation. The expression of *E. sinensis* Peroxiredoxin4 (*EsPrx4*) was detected when the most stable and least stable internal reference genes were selected, respectively. The results showed that reference genes with different stability had great influence on the accurate results of the target gene expression. In the Chinese mitten crab (*E. sinensis*), AK and EF-1α were the most suitable reference genes under the stimulation of *V. anguillarum*. Under the stimulation of copper ions, *GAPDH* and *β-ACTIN* were the most suitable reference genes. This study provided important information for further research on immune genes in *V. anguillarum* or copper ion stimulation.

## 1. Introduction

Quantitative real-time polymerase chain reaction (qRT-PCR) has become the most frequently used technique for analyzing gene expression with the advantages of high sensitivity, strong specificity and simple operation [1]. In qRT-PCR experiments, appropriate reference genes are usually selected to reduce experimental errors and to ensure the reliability of the results [2,3]. Generally, the ideal reference gene should have relatively stable expression levels in different tissues and cells, in different cell cycles and at different development stages and under various experimental conditions [4]. Many studies have shown that no reference gene is suitable for all experimental conditions [5,6]. Therefore, it is important to screen and evaluate appropriate reference genes under experimental conditions. There are many commonly used reference genes, such as glyceraldehyde-3-phosphate dehydrogenase (*GAPDH*), β-actin (*β-ACTIN*), elongation factor 1α (*EF-1α*), 18S ribosomal (*18S*), α-tubulin (*α-TUB*), etc. [7,8,9]. At present, there are many reports of reference genes for aquatic animals, such as *Crassostrea nippona* [10], *E. sinensis* [11], *Penaeus monodon* [12], *Ostrea edulis* [13] and *Penaeus japonicus* [14].

Four kinds of software are usually used to evaluate the stability of reference genes: geNorm (V3.5, Ghent, Belgium) [15], NormFinder (V0953, Aarhus, Denmark) [16], BestKeeper (V ersion1.0, Munich, Germany) [17] and Ref-Finder (Version 1.0, http://www.leonxie.com/referencegene.php (accessed on 4 May 2021)). geNorm selects more stable reference genes by calculating the average expression stability value (M), which is defined as the average pairwise variation of the particular gene with all other control genes. The criterion is that the smaller the M value, the more stable the reference gene. To determine the optimal number of reference genes, the pairwise variation (V) between sequential ranked genes (V_n_/V_n+1_) is calculated using geNorm. If the value of V_n_/V_n+1_ is <0.15, the optimal number of reference genes is n; when the value of V_n_/V_n+1_ is >0.15, the best reference gene number is n + 1. The principle of calculation and criterion of NormFinder is similar to that of geNorm. The stable value of reference gene expression is obtained first, and then the optimal reference gene is selected. The lower the stability value, the better the stability of expression. BestKeeper determines the reference gene by comparing standard deviation (SD). The smaller the standard deviation and coefficient of variation, the more stable the reference gene. Ref-Finder is applied to determine the comprehensive ranking of the most stable reference genes by geNorm, NormFinder and BestKeeper. Many studies have shown that it is not recommended to use only one reference gene, but to select multiple reference genes, which could improve the accuracy of gene expression analysis [18,19,20].

The Chinese mitten crab (*E. sinensis*), commonly known as the river crab and hairy crab, belongs to the phylum Arthropoda, subphylum Crustacea, class Malacostraca, order Decapoda, family Varunidae, genus *Eriocheir*. It is also named for the dense fluff on its claws. The Chinese mitten crab is a traditional aquatic treasure in China [21]. It is widely distributed and can be found in many rivers and lakes in China. It is one of the most important farmed crab species in China and has a wide market demand, with important economic and scientific value. Since China began vigorously promoting the cultivation of river crabs, the total annual production of river crabs has been approximately 750,000 tons every year, and the national production of river crab cultivation in 2018 was 757,000 tons, an increase of 0.79% over 2017. The industry is expected to be worth approximately CNY 1.25 billion annually [22].

The Chinese mitten crab (*E. sinensis*) is an important economic aquatic animal in China. At present, more and more studies on *E. sinensis* are concerned with the aspects of cultivation, reproductive development, disease and nutrition [23,24]. Some of these studies involved gene expression analysis. In order to obtain reliable results, it was necessary to screen stable reference genes. Huang et al. [11] compared the expression level of 10 reference genes in different tissues and different development and molting stages of *E. sinensis*. The result showed that the expression level of reference genes was not stable, which implied that the stability of the reference gene should be assessed to screen out the suitable reference gene depending on different experimental conditions.

Two experimental conditions were chosen for this study. As Chinese mitten crab (*E. sinensis*) production increases, diseases are becoming increasingly serious, especially bacteria, which seriously affect the economic returns of Chinese mitten crab (*E. sinensis*) farming. There are many bacterial diseases related to the Chinese mitten crab, the main causative agent being *Vibrio*, which causes mostly localized infections and then transforms into systemic septicemia [25]. Therefore, the first experimental condition was stimulation by *V. anguillarum*, which can cause hemorrhagic septicemia in crustaceans and is one of the pathogens with a high lethality rate in crustaceans [26]. Copper sulphate is often used to prevent moss and cyanobacterial outbreaks in crustacean production. Copper ions are essential for the growth and development of crustaceans as they are responsible for metabolism, growth and development and immune regulation, and have an important influence on hematopoiesis, cell reproduction and enzyme activity [27,28]. However, excessive amounts of copper ions can cause oxidative damage to various organs and tissues of aquatic organisms. So, the second experimental condition was copper ion stress, which has been shown to cause a decrease in crustacean immunity and, subsequently, increase pathogen susceptibility, leading to disease outbreaks during breeding [29,30,31]. Both of these experimental conditions can cause oxidative stress in Chinese mitten crabs. Peroxiredoxin (*Prx*) is a superfamily of nonselenium peroxidases which has been demonstrated to play important roles in reducing and detoxifying hydrogen peroxide [32,33], peroxynitrite and a wide range of organic hydroperoxides (*ROOH*) [34]. *EsPrx4* belongs to the superfamily of antioxidant proteins and plays an important role in innate immunity. The only protein in the *Prx* family with a secreted signal peptide, *Prx4* was first discovered to act as an antiviral agent for DAMPs. The expression of *Prx4* was found to be significantly up-regulated and the secretion of extracellular *Prx4* was increased after white spot syndrome virus (WSSV) infection in *Fenneropenaeus chinensis* [35]. In this study, the expression level of 10 candidate reference genes including *AK*, *UBE*, *GST*, *GAPDH*, *EF-1α*, *α-TUB*, *HSP90*, *β-ACTIN*, *EF-2* and *PGM2* was analyzed by qRT-PCR in hepatopancreas tissue of *E. sinensis* stimulated by *Vibrio anguillarum* and copper ions. The geNorm, NormFinder, BestKeeper and Ref-Finder software were used to evaluate the stability of the 10 reference genes, which was further verified by the expression pattern of the *E. sinensis* Peroxiredoxin4 (*EsPrx4*) gene. *EsPrx4* belongs to the superfamily of antioxidant proteins and plays an important role in innate immunity. This work provides a piece of valuable information related to appropriate reference gene selection in *E. sinensis*.

## 2. Material and Methods

### 2.1. Animals

The crabs were collected from Yangcheng Lake, Suzhou City, Jiangsu Province, with an average weight of 78 ± 5 g, and kept in the laboratory for one week. The water temperature during temporary culture was 27 ± 1 °C and the dissolved oxygen level was maintained at 7 ± 0.2 mg/L, and the river crabs were fed with commercial feed at 5 pm daily.

### 2.2. Vibrio anguillarum and Copper Ion Stimulation

The *V. anguillarum* used in the experiments was provided by the Freshwater Fisheries Research Centre of the Chinese Academy of Fisheries Sciences. *V. anguillarum* was diluted to 1 × 10^7^ CFU/mL by PBS buffer and 40 μL *V. anguillarum* was injected separately into the connection between the third foot and the body cavity of the river crab. Hepatopancreas tissues were taken at 0 h (Control group), 6 h, 12 h, 24 h, 48 h and 72 h in each group, with three biological replicates for each time point. The experimental concentration gradient was set according to the semi-lethal concentration of copper ions, on the basis of the lethal concentration 50 (LC50) of 22.16 mg/L. The experiment set up a blank control group (0 mg/L) and four Cu^2+^ concentration treatment groups: 11.08 mg/L, 2.77 mg/L, 0.69 mg/L and 0.17 mg/L (1/2 LC_50_, 1/8 LC_50_, 1/32 LC_50_ and 1/128 LC_50_). Hepatopancreas tissues were taken after 10 days of stimulation, with three biological replicates for each group. Hepatopancreas tissue was quickly obtained by dissecting river crabs on ice and immediately submerged completely in RNA keeper (Vazyme, Nanjing, China). The submerged tissue was refrigerated overnight at 4 °C and stored at −80 °C for further extraction of total RNA.

### 2.3. RNA Extraction and Reverse Transcription (RT)

Total RNA was extracted using the TRIzol reagent (Vazyme, Nanjing, China) according to the manufacturer’s instructions; RNA quality and concentration were determined by a spectrophotometer. RNA with an OD260/280 ratio between 1.9 and ~2.2 was considered satisfactory and was used for subsequent analysis. For RT-PCR, 1 μg RNA was converted to cDNA with a total reaction volume of 20 μL using the HiScript II Q RT Supermix for qPCR (+gDNA wiper) Kit (Vazyme) according to the manufacturer’s protocol. Reverse transcription was performed using the HiScript II Q RT SuperMix for qPCR (+gDNA wiper) kit (Vazyme). An amount of 1 μg RNA was used in the reverse transcription system for each sample stimulated by *V. anguillarum* and copper ions in a total reaction volume of 20 μL. In the first step, genomic DNA was removed and the reaction consisted of: 4 μL 4× gDNA wiper Mix, 1 μg RNA, RNase-free ddH_2_O added to 16 μL, 42 °C for 2 min; the second step was a reverse transcription system: 5× Hiscript III qRT SuperMix 4 μL with the reaction solution from the first step, 37 °C for 15 min, 85 °C for 5 s. The product was used immediately for subsequent experiments.

### 2.4. Primer Specificity and Amplification Efficiency

Ten candidate reference genes including *β-ACTIN*, *EF-1α*, *EF-2*, *GAPDH*, *α-TUB*, *AK*, *UBE*, *GST*, *HSP90* and *PMG2* were selected. Gene sequences were taken from the transcriptome of *E. sinensis* (data unpublished). Primers were designed by Primer 5 (Premier, Winnipeg, MB, Canada) and synthesized by Shanghai Tianlin Biotechnology Company Limited (Table 1). The amplification specificity of the primers was evaluated using 2.0% agarose gel electrophoresis and melting curve. The amplification efficiency of the primers was calculated according to the standard curve, which was generated by 0, 5-, 10-, 100- and 1000-times dilutions series of hepatopancreatic cDNA. A correlation coefficient (R^2^) value above 0.98 was accepted.

### 2.5. Relative Quantification of Gene Expression

Nanjing Vazyme ChamQ SYBR qPCR Master mix was used in fluorescence quantitative detection. The 20 μL reaction system consisted of 10 μL Master mix, 2 μL cDNA sample, 1 μL upstream and downstream primers of 10 μM, respectively, and ddH_2_O replenished to 20 µL reaction system. The reaction procedure was as follows: 95 °C for 30 s, 40 cycles of the following denaturation at 95 °C for 5 s and renaturation at 60 °C for 30 s. A melting curve was generated by the following procedure, gradually increasing amplification temperature at a rate of 0.2 °C from 65 °C to 92 °C.

### 2.6. Data Analysis

According to the results of fluorescence quantitative analysis, the stability of 10 candidate reference genes was assessed. The geNorm algorithm first found the minimum Ct value from all samples, and then subtracted the minimum Ct value from the Ct values of the remaining sample, thus obtaining the ΔCt value. Then, the 2^−ΔCt^ value of the corresponding gene was calculated. The NormFinder algorithm was the same as that of geNorm, both of which obtained the 2^−ΔCt^ value of the corresponding gene. The BestKeeper algorithm directly input the Ct value into the Excel table with the built-in formula, and automatically calculated the equivalent of correlation coefficient (r), standard deviation (SD) and coefficient of variation (CV). The Ref-Finder algorithm was a synthesis of geNorm, NormFinder and BestKeeper to eliminate the one-sidedness of the independent use of three kinds of software. To verify the reference gene effectiveness, two reference genes with the best stability and one with the worst stability were chosen to calculate the expression of the *EsPrx4* gene by the 2^−ΔΔCt^ method [36]. *EsPrx4* information is shown in Table 1.

## 3. Results and Analysis

### 3.1. Specificity and Amplification Efficiency of Primers

In order to improve the accuracy of qRT-PCR results, primers should have high specificity and amplification efficiency [36]. In this study, the amplification efficiencies of all primer pairs were above 95.7%, and R^2^ values ranged from 0.9908 to 0.9986. Ten candidate reference genes were amplified by RT-PCR, and the products were shown as a single band of the expected size (Figure 1). The dissociation curves of ten reference genes showed no primer dimer and non-specific amplification, which further proved the good specificity of the primers (Figure 2).

### 3.2. Ct Values of 10 Candidate Reference Genes

The average Ct values of the 10 candidate internal reference genes under the *V. anguillarum* stimulation ranged from 19.34 to 29.88, with the highest expression being that of *EF-1α*. The expression of the 10 candidate internal reference genes under copper ion stimulation ranged from 21.29 to 29.05, with *AK* having the highest expression (Figure 3).

### 3.3. Stability of 10 Candidate Reference Genes

geNorm was used to analyze the stability of the 10 candidate reference genes. The geNorm software can be used to calculate the average expression stability (M) of the reference genes and rank them to find the most suitable reference genes, where a smaller M value means a more stable gene. As shown in Figure 4A, the stability order of the 10 candidate reference genes under the stimulation of *V. anguillarum* was *AK*|*EF-1α* > *UBE* > *GAPDH* > *α-TUB* > *EF-2* > *PGM2* > *GST* > *β-ACTIN* > *HSP90*. *AK* and *EF-1α* were the most stable reference genes, and HSP90 was the most unstable. As shown in Figure 4B, the stability order of the 10 candidate reference genes under copper ion stimulation was as follows: *β-ACTIN*|*GAPDH* > *EF-1α > EF-2 > α-TUB > GST > PGM2 > AK > HSP90. β-ACTIN* and *GAPDH* were the most stable reference genes, while *HSP90* was the most unstable.

In studies requiring high accuracy in expression quantification, the use of a single internal reference gene is not sufficient for experimental purposes. Therefore, the use of two or more stably expressed internal reference genes to correct for target gene expression has become a new guideline in quantitative PCR [37], and Vandesompele et al. [15] suggest that when V is less than 0.15, there is no need to add more internal reference genes for correction. In this experiment, the paired variations (V) of each internal reference gene was also analyzed (Figure 5). As shown in Figure 5A, the value of V2/3 was <0.15 under *V. anguillarum* stimulation, indicating that two reference genes could be accurately standardized (Figure 5A). However, the V9/10 value was >0.15 under copper ion stimulation, indicating that more reference genes were needed for proofreading (Figure 5B).

NormFinder analysis showed that the most stable candidate reference gene stimulated by *V. anguillarum* was *AK*, with a stability M value of 0.5. However, the most unstable gene was *HSP90*, with a stability M value of 1.714. The most stable gene under copper stimulation was *GAPDH*, with a stability M value of 0.334. However, the most unstable reference gene was *HSP90*, with a stability value of 1.589 (Table 2).

The BestKeeper software evaluates the expression stability of candidate reference genes based on standard deviation (SD) and coefficient of variation (CV), where the smaller the SD and CV, the higher the stability of the reference genes. BestKeeper analysis showed that the most stable reference gene stimulated by *V. anguillarum* was *α-TUB* with an SD value of 1.11, and the most unstable reference gene was *PGM2* with an SD value of 2.15. The most stable reference gene stimulated by copper ions was *PGM2* with an SD value of 0.58, and the most unstable reference gene was *UBE* with an SD value of 1.76 (Table 3).

The Ref-Finder web-based tool was applied to comprehensively analyze the 10 candidate reference genes. Stimulated by *V. anguillarum*, the most stable reference gene was *AK*, followed by *EF-1α,* and the most unstable reference gene was *HSP90* (Figure 6A). Stimulated by copper ions, the most stable reference gene was *GAPDH*, and the most unstable gene was *HSP90* (Figure 6B).

### 3.4. Validation of Reference Genes

In order to verify the reliability of the appropriate reference gene, *EsPrx4* expression stimulated by *V. anguillarum* and copper ions was evaluated. Under the stimulation of *V. anguillarum*, the expression trend of *EsPrx4* was similar when the two most stable reference genes (*AK* and *EF-1α*) were selected. However, the expression level of *EsPrx4* was totally changed, especially at 24 h, when the least stable reference, *HSP90,* was selected. It was 3.18 times and 3.03 times higher than that of *AK* and *EF-1α* as the reference genes, respectively (Figure 7A). Under the stimulation of copper ions, the expression trend of *EsPrx4* was similar when the two most stable reference genes (*GAPDH* and *β-ACTIN*) were selected. When *HSP90* (the least stable reference) was selected as the reference gene, the expression level of *EsPrx4* in the 0.69 mg/L group was 5.34 times and 4.18 times higher than that of *GAPDH* and *β-ACTIN* as the reference genes, respectively (Figure 7B).

## 4. Discussion

In order to obtain the relative expression level of the target genes more reliably, it is necessary to screen the suitable reference gene according to different experimental conditions. *EF-1α, GAPDH* and *β-ACTIN* have always been regarded as the most stable reference genes. However, many studies had revealed that *GAPDH* and *β-actin* were not stably expressed under different conditions and could not be directly selected as reference genes [38,39,40,41]. This was consistent with our findings. We evaluated the stability of 10 candidate reference genes, analyzed the optimal number and verification of reference genes under different experimental conditions and confirmed the importance of screening reference genes.

geNorm, NormFinder, BestKeeper and Ref-Finder were applied to evaluate the stability of the reference gene, but different program algorithms could lead to slightly different stability [42]. In this study, under the stimulation of *V. anguillarum*, *AK* and *EF-1α* were the most stable genes based on the analysis of geNorm and NormFinder, and *α-TUB* was the most stable gene based on the analysis of BestKeeper. Under the stimulation of copper ions, *GAPDH* and *β-ACTIN* were the most stable genes based on the analysis of geNorm and NormFinder, and *PGM2* was the most stable gene based on the analysis of BestKeeper. To solve this inconsistent result, Ref-Finder was used to conduct a comprehensive analysis. The result showed that *AK* was the most stable reference gene stimulated by *V. anguillarum*, and *GAPDH* was the most stable reference gene stimulated by copper ions. *HSP90* was the most unstable reference gene stimulated by *V. anguillarum* and copper ions. Furthermore, *EsPrx4* expression was evaluated under the stimulation of *V. anguillarum* and copper ions when the most and the least stable reference genes were selected. The results showed that *EsPrx4* expression fluctuated, which implied that the target gene’s accurate expression level depended on the appropriate reference gene.

Similarly, the reference gene stability of *Penaeus stylirostris* was analyzed under the infection of WSSV, which indicated that *EF-1α* and *GAPDH* were the most stable references [19]. In *Portunus trituberculatus*, it was found that the stability of reference genes in different tissues after WSSV challenge was different. *GAPDH* and cyclophilin A were the most stable reference genes in gills and hemocytes, respectively [38]. In the estuary mud crab (*Scylla paramamosain*), *18s* and *β-actin* were the most stable reference genes when it was exposed to heavy metal conditions [43]. The results of a study of six different developmental stages of the ovaries of *Procambarus clarkii* showed that *EF-1α* was the most consistently expressed of the endogenous genes [44]. These results implied that the reference genes’ stability fluctuated under different experimental conditions. Therefore, we should select different reference genes according to the species, tissue type and experimental conditions.

In conclusion, we compared the stability of 10 candidate reference genes under two different experimental conditions. The results showed that reference genes with different stability had great influence on the accurate results of the target gene expression. Under the stimulation of *V. anguillarum*, *AK* and *EF-1α* were the most suitable reference genes. Under the stimulation of copper ions, *GAPDH* and *β-ACTIN* were the most suitable reference genes. At the same time, two stable reference genes were suggested as selections for obtaining accurate gene expression data. This work will provide useful information in *E. sinensis* gene expression studies.

## Figures and Tables

**Figure 1 genes-14-01099-f001:**
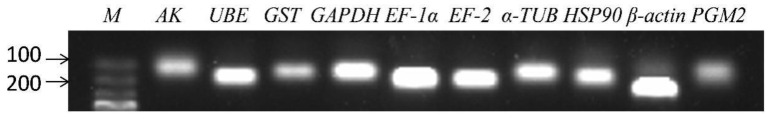
Agarose gel electrophoresis of fragments of 10 candidate reference genes. The “M” in this figure is the marker. The genes in this figure include arginine kinase (*AK*), ubiquitin-conjugating enzyme E2b (*UBE*), glutathione S-transferase (*GST*), glyceraldehyde-3-phosphate dehydrogenase (*GAPDH*), elongation factor 1α (*EF-1α*), elongation factor 2 (*EF-2*), α-tubulin (*α-TUB*), heat shock protein 90 (*HSP90*), β-actin (*β-ACTIN*), phosphoglucomutase 2 (*PGM2*).

**Figure 2 genes-14-01099-f002:**
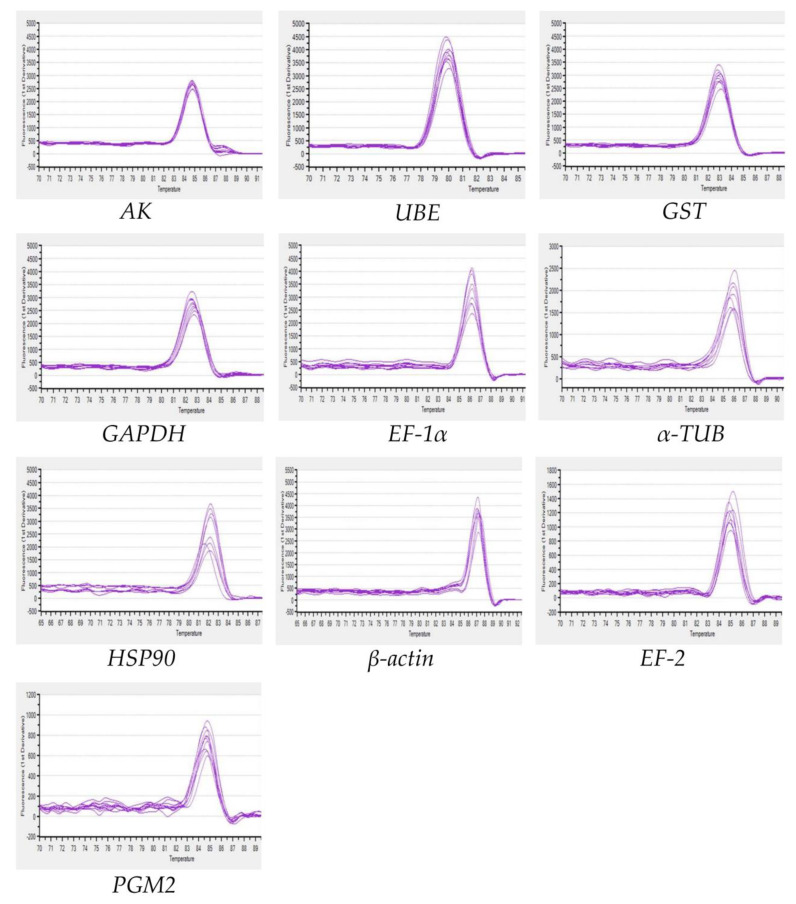
Dissociation curves of 10 candidate reference genes. The genes in this figure include arginine kinase (*AK*), ubiquitin-conjugating enzyme E2b (*UBE*), glutathione S-transferase (*GST*), glyceraldehyde-3-phosphate dehydrogenase (*GAPDH*), elongation factor 1α (*EF-1α*), elongation factor 2 (*EF-2*), α-tubulin (*α-TUB*), heat shock protein 90 (*HSP90*), β-actin (*β-ACTIN*), phosphoglucomutase 2 (*PGM2*).

**Figure 3 genes-14-01099-f003:**
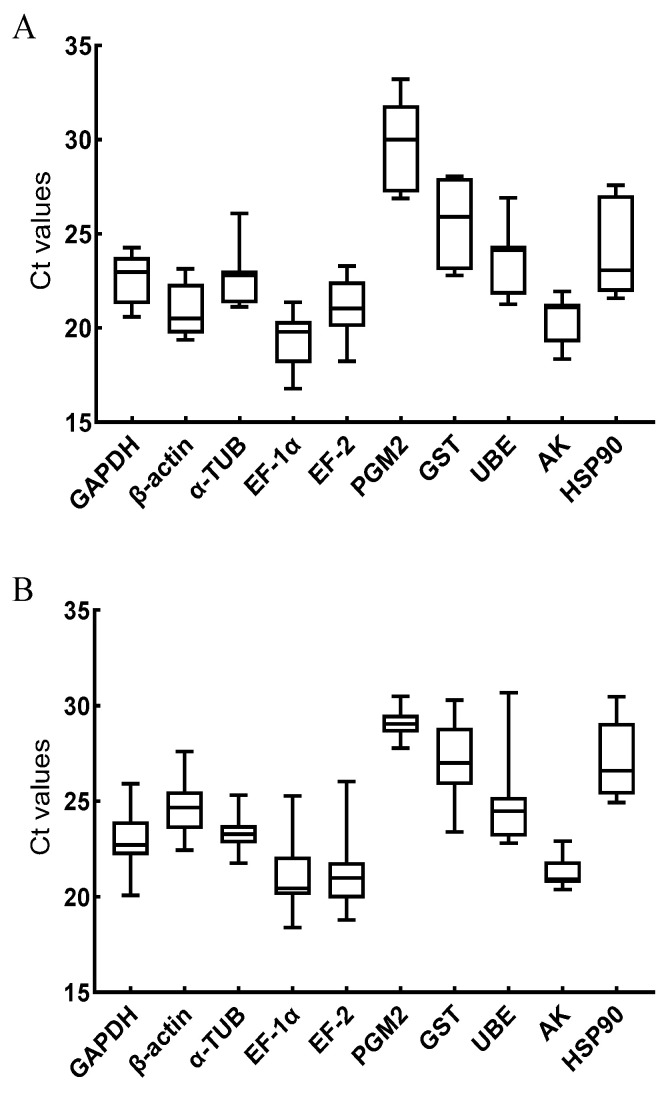
The average Ct values of 10 candidate reference genes stimulated by *V. anguillarum* (**A**) and copper ions (**B**). The genes in this figure include glyceraldehyde-3-phosphate dehydrogenase (*GAPDH*), β-actin (*β-ACTIN*), α-tubulin (*α-TUB*), elongation factor 1α (*EF-1α*), elongation factor 2 (*EF-2*), phosphoglucomutase 2 (*PGM2*), glutathione S-transferase (*GST*), ubiquitin-conjugating enzyme E2b (*UBE*), arginine kinase (*AK*), heat shock protein 90 (*HSP90*).

**Figure 4 genes-14-01099-f004:**
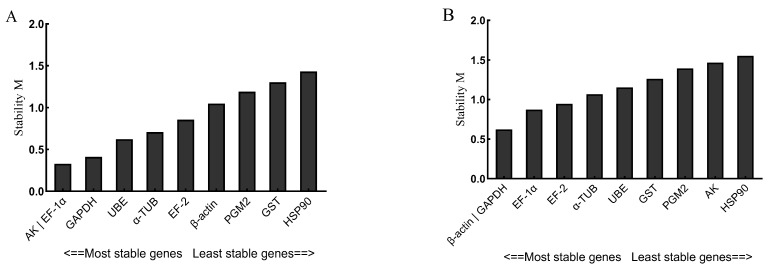
The stability M values evaluated by geNorm of 10 candidate reference genes stimulated by *V. anguillarum* (**A**) and copper ions (**B**). The genes in this figure include arginine kinase (*AK*), elongation factor 1α (*EF-1α*), glyceraldehyde-3-phosphate dehydrogenase (*GAPDH*), ubiquitin-conjugating enzyme E2b (*UBE*), α-tubulin (*α-TUB*), elongation factor 2 (*EF-2*), β-actin (*β-ACTIN*), phosphoglucomutase 2 (*PGM2*), glutathione S-transferase (*GST*), heat shock protein 90 (*HSP90*). The vertical coordinate is the average expression stability value “M”.

**Figure 5 genes-14-01099-f005:**
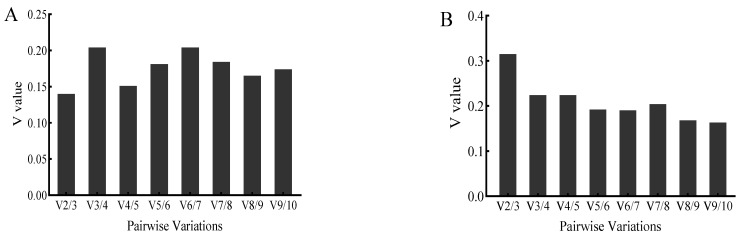
Paired variations of 10 candidate reference genes stimulated by *V. anguillarum* (**A**) and copper ions (**B**). In this figure, the horizontal coordinate indicates the optimal number of reference genes, and the vertical coordinate indicates the pairwise variation V.

**Figure 6 genes-14-01099-f006:**
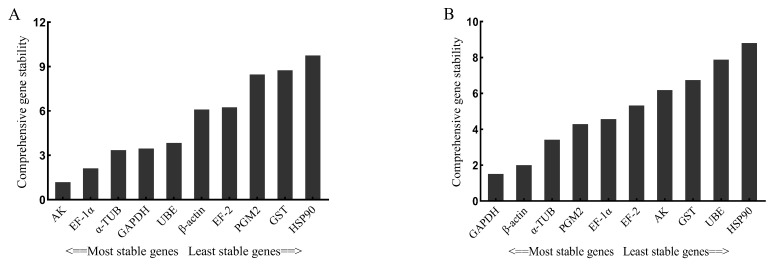
Comprehensive analysis by Ref-Finder of 10 candidate reference genes stimulated by *V. anguillarum* (**A**) and copper ions (**B**). The genes in this figure include arginine kinase (*AK*), elongation factor 1α (*EF-1α*), α-tubulin (*α-TUB*), glyceraldehyde-3-phosphate dehydrogenase (*GAPDH*), ubiquitin-conjugating enzyme E2b (*UBE*), β-actin (*β-ACTIN*), elongation factor 2 (*EF-2*), phosphoglucomutase 2 (*PGM2*), glutathione S-transferase (*GST*), heat shock protein 90 (*HSP90*).

**Figure 7 genes-14-01099-f007:**
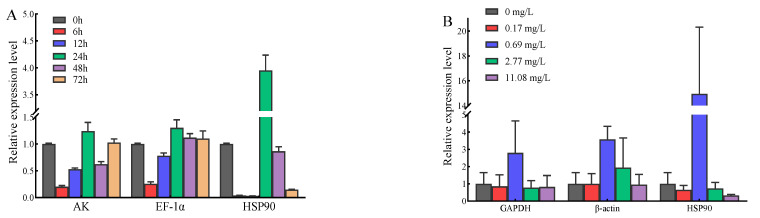
Relative expression of *EsPrx4* stimulated by *V. anguillarum* (**A**) and copper ions (**B**) based on different reference genes. The genes in (**A**) include arginine kinase (*AK*), elongation factor 1α (*EF-1α*), heat shock protein 90 (*HSP90*). The genes in (**B**) include glyceraldehyde-3-phosphate dehydrogenase (*GAPDH*), β-actin (*β-ACTIN*), heat shock protein 90 (*HSP90*).

**Table 1 genes-14-01099-t001:** Ten candidate reference genes and *E. sinensis* Peroxiredoxin4 (*EsPrx4*) used in this study.

Gene	Primer Sequences (5′-3′)	Product Size (bp)	Correlation Coefficient (R^2^)	Amplification Efficiency (%)
*β-ACTIN*	F:GCATCCACGAGACCACTTACA R:CTCCTGCTTGCTGATCCACATC	266	0.9937	106.7%
*EF-1α*	F:AGGTCGGCTACAACCCAACT R:TGAACTCGTAGGAGCCACTC	138	0.9908	104%
*EF-2*	F:TGATGGGTCGCTTTGTTGAG R:GGTCAGATGGGTTCTTGGGT	192	0.9908	104.3%
*HSP90*	F:TCACCAACGACTGGGAGGAT R:CAGGAAGAGGAGTGCCCTGA	83	0.9918	104.5%
*AK*	F:GCTCAAGGCCAAGAAGACCA R:CATACACACCGACGCCAGAG	92	0.9947	108.1%
*UBE*	F:TTGCGTTCACAACTCGTATCTACC R:GTCCGTGAGGAGGGAACAGA	137	0.9927	107.3%
*GST*	F:GCTGTGGTGGAGCGACTCA R:TCCAACTCCTCTCCACGGAA	98	0.9923	95.7%
*GAPDH*	F:GCGTGTTCACCACCATTGAG R:ACATGGGTGCATCAGCAGAG	93	0.9979	100%
*α-TUB*	F:GTGGAGATCTGGCCAAGGTG R:CCCACATACCAGTGCACGAA	136	0.9986	101%
*PGM2*	F:CTGACGGGCTTCAAGTGGAT R:TCCTTGTCCAACACCTCCGA	122	0.9931	107.2%
*EsPrx4*	F:CACGAAAGGAAGGTGGACTG R:TCATCCACAGACCTTCCGACT	193	0.9973	107.7%

The genes in this table include β-actin (*β-ACTIN*), elongation factor 1α (*EF-1α*), elongation factor 2 (*EF-2*), heat shock protein 90 (*HSP90*), arginine kinase (*AK*), ubiquitin-conjugating enzyme E2b (*UBE*), glutathione S-transferase (*GST*), glyceraldehyde-3-phosphate dehydrogenase (*GAPDH*), α-tubulin (*α-TUB*), phosphoglucomutase 2 (*PGM2*), *E. sinensis* Peroxiredoxin4 (*EsPrx4*).

**Table 2 genes-14-01099-t002:** Stability values evaluated by NormFinder of 10 candidate reference genes stimulated by *V. anguillarum* and copper ions.

Gene Name	*Vibrio anguillarum*	Copper Ions
*EF-1α*	0.574	1.056
*AK*	0.500	1.465
*EF-2*	1.133	1.094
*UBE*	0.590	1.381
*β-ACTIN*	1.212	0.436
*GST*	1.372	1.286
*GAPDH*	0.737	0.334
*α-TUB*	0.833	0.661
*HSP90*	1.714	1.589
*PGM2*	1.368	1.346

The genes in this table include elongation factor 1α (*EF-1α*), arginine kinase (*AK*), elongation factor 2 (*EF-2*), ubiquitin-conjugating enzyme E2b (*UBE*), β-actin (*β-ACTIN*), glutathione S-transferase (*GST*), glyceraldehyde-3-phosphate dehydrogenase (*GAPDH*), α-tubulin (*α-TUB*), heat shock protein 90 (*HSP90*), phosphoglucomutase 2 (*PGM2*).

**Table 3 genes-14-01099-t003:** Stability values evaluated by BestKeeper of 10 candidate reference genes stimulated by *V. anguillarum* and copper ions.

Gene Name	*Vibrio anguillarum*		Copper Ion	
	SD (±Ct)	CV (%Ct)	SD (±Ct)	CV (%Ct)
*AK*	1.12	5.48	0.67	3.15
*UBE*	1.48	6.22	1.76	6.97
*GST*	2.05	7.98	1.54	5.71
*GAPDH*	1.13	4.98	1.19	5.16
*EF-1α*	1.23	6.34	1.66	7.78
*α-TUB*	1.11	4.86	0.79	3.38
*HSP90*	2.14	8.89	1.51	5.57
*β-ACTIN*	1.16	5.52	1.14	4.64
*EF-2*	1.56	7.4	1.65	7.71
*PGM2*	2.15	7.2	0.58	1.98

The genes in this table include arginine kinase (*AK*), ubiquitin-conjugating enzyme E2b (*UBE*), glutathione S-transferase (*GST*), glyceraldehyde-3-phosphate dehydrogenase (*GAPDH*), elongation factor 1α (*EF-1α*), α-tubulin (*α-TUB*), heat shock protein 90 (*HSP90*), β-actin (*β-ACTIN*), elongation factor 2 (*EF-2*), phosphoglucomutase 2 (*PGM2*). The values in this table are the standard deviation (SD) and coefficient of variation (CV) of the Ct values of candidate reference genes.

## Data Availability

All data generated or analyzed during this study were included in this article.

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
