# Peer review of "Screening of Suitable Reference Genes for Immune Gene Expression Analysis Stimulated by *Vibrio anguillarum* and Copper Ions in Chinese Mitten Crab (*Eriocheir sinensis*)"

_genes, 2023, doi:10.3390/genes14051099_

Round 1

Reviewer 1 Report

I appreciate the authors for making an attempt to select a suitable reference gene for expression studies for different conditions. The selection of reference genes is still debatable when it comes to varying experimental conditions. As far as I am concerned, the manuscript is well structured and the results were presented clearly. But I would suggest the authors to consider some points discussed below and revise the manuscript accordingly.

The authors should mention in the introduction or elsewhere why they have chosen these specific two experimental conditions (V. anguillarum and copper ions) and their impact on selected species

From the Line 53-72, the authors speak about the software but they have written in past tense format. The software is still used by many so the authors need to change the grammar (present tense).

If the authors use any specific software, it is always better to use the company and country name or the website link in brackets (e.g. for Primer 5). In addition if the authors use instrument, please follow the same (e.g. spectrophotometer).

The methodology section should have all the details and it should be replicable so that the other researchers will follow your methods to have similar results without any hindrances. So try to elaborate the experiment groups and their conditions (section 2.2).

When you come to the discussion part, it would be scientifically good if you give some justification sentences with recent citations why some genes are stable and why some are not. The manuscript deals with the experimental animal crab so discuss with crustacean species is fine but comparing with the finfish is not reliable. So try to avoid the citations dealing with fish e.g “Schizothorax prenanti”.    

I have given some comments below for the authors to make this manuscript scientifically sound. But the authors should check throughout the manuscript and avoid these minor errors.

Abstract

Line 14 – Try to avoid abbreviations in abstract unless you mention again elsewhere in the abstract again. So expand “qRT-PCR”

Line 20, Instead of saying “Their expression level” I would suggest to use “Expression levels of these reference genes were”

Line 27 – Genus species name should be in italics. please try to be consistent with italics throughout the manuscript.

Line 28 – “The results….” is not clearly manifest the research work done in this manuscript, it would be better if you revise the line.

Line 29-31 – Include the experimental animal and be specific.

Introduction section

Line 38 – “appropriate reference genes were” it is still followed so write it in present tense.

Line 37- 40 are not clear, rewrite the sentence

Line 40-41 Rather than using the word “different” like “different tissues and cells, in different cell cycles and different development stages, and under different experimental conditions” just combine the terms and make it easy for the readers

Line 59 – the word “number of” has been repeated. Please remove

Line 69-71, the author has indicated “many studies” but cited only one reference. The sentence should be justified by giving 2 or more citations.

Line 74-76 – Cite suitable references for this sentence.

Line 78 – change to “different development and molting stages”

Materials and methods

Section 2.1

Line 94-96  “The crab was taken” but I would prefer to write “The crabs were collected” and please indicate the city where the lake is located. The authors indicated the average weight of the crabs but mentioned size so replace the word “size” with “average weight”. Write as “The water temperature was maintained at 27 ± 1°C”. The crabs were fed with what? Please indicate that.

Section 2.2

give the reference or source of V. anguillarum. Give more details about the group and the control group (e.g. no control group was mentioned for V. anguillarum but control was mentioned for copper ion). Where did you injected in the crab. The authors should mention that.

Section 2.5

20 μL ddH2O? for 20 μL reaction volume. Please check and revise.

Section 2.6

Line 131-133. The line is confusing “all samples” and “other samples”. What these samples indicate control or treated? Revise the line or better to give it in formula.

Change the Table 1 legend and specifically mention the primer details. Authors mentioned genes but the table contains the details of primers.

Results

Section 3.2

Line 158-160, the sentence is not clear and revise the line and explain the results legibly.

In Table 2 – authors indicate the best reference gene in the last row. But I would suggest to remove from the table because it was already mentioned in the results section. Leave the table as it shows only the stability value. Remove the brackets from V. anguillarum and copper ions.

Figure 6 - Change "Comprehensively" to "Comprehensive"

Reviewer 2 Report

This topic is of high interest to those who work with animals whose life stages differ dramatically from one another, such as with molting animals with much cell turnover.  This fact is understated in the manuscript - see Line 224 with only two citations.  The authors need to find more examples as justification for the topic.  This reviewer provided one such reference.  The choice of species needs a better rationale, as does the target gene, and the reasons behind the specific pathogen and element used as stimulants. Include what the stimulants are expected to do with the target gene and the reference genes - and perhaps why the similarity or difference in expression.  Expound a bit more on mechanism.  

If the main point of the paper was to compare the three analysis software packages, then bring that forward more.  If the main point is to focus on what happens to this important target gene (explain it in a thorough sense), then bring that forward.

Several marks are on the paper and I hope they can be read as they are light on the page.  Constructive points are on the paper and some here:

The figures are too small, all genes need to be listed in each figure/table legend (footnote, at least). 

Is there a standard way to write the genes - as in italics?

English, verb tense, and grammar needs to be revisited.  

If there was only one crab (line 94-5), then the paper should not be published.  A sample size of one is not acceptable.  However, later there are plurals.  It is confusing.

Round 2

Reviewer 2 Report

The manuscript is improved, however some of the original comments provided by this reviewer were disregarded.  In particular, that each table and figure must be able to stand alone (include descriptions of all abbreviations, for instance) and to provide citations in the correct places (e.g., line 224) that many studies revealed GAPDH and actin could not be selected as reference genes.  Two studies is not "many".  Please go back and address the suggestions that were passed over. 
